# Learning Reasoning Paths over Semantic Graphs for Video-grounded Dialogues

**Hung Le**[†‡]**, Nancy F. Chen**[‡]**, Steven C.H. Hoi**[†§]
† Singapore Management University
`hungle.2018@smu.edu.sg`
‡ A*STAR, Institute for Infocomm Research
`nfychen@i2r.a-star.edu.sg`
§ Salesforce Research Asia
`shoi@salesforce.com`

## Abstract

Compared to traditional visual question answering, video-grounded dialogues require additional reasoning over dialogue context to answer questions in a multi-turn setting. Previous approaches to video-grounded dialogues mostly use dialogue context as a simple text input without modelling the inherent information flows at the turn level. In this paper, we propose a novel framework of Reasoning Paths in Dialogue Context (PDC). PDC model discovers information flows among dialogue turns through a semantic graph constructed based on lexical components in each question and answer. PDC model then learns to predict reasoning paths over this semantic graph. Our path prediction model predicts a path from the current turn through past dialogue turns that contain additional visual cues to answer the current question. Our reasoning model sequentially processes both visual and textual information through this reasoning path and the propagated features are used to generate the answer. Our experimental results demonstrate the effectiveness of our method and provide additional insights on how models use semantic dependencies in a dialogue context to retrieve visual cues.

## 1 Introduction

Traditional visual question answering (Antol et al., 2015; Jang et al., 2017) involves answering questions about a given image. Extending from this line of research, recently Das et al. (2017); Alamri et al. (2019) add another level of complexity by positioning each question and answer pair in a multi-turn or conversational setting (See Figure 1 for an example). This line of research has promising applications to improve virtual intelligent assistants in multi-modal scenarios (e.g. assistants for people with visual impairment). Most state-of-the-part approaches in this line of research (Kang et al., 2019; Schwartz et al., 2019b; Le et al., 2019) tackle the additional complexity in the multi-turn setting by learning to process dialogue context sequentially turn by turn. Despite the success of these approaches, they often fail to exploit the dependencies between dialogue turns of long distance, e.g. the $2^{nd}$ and $5^{th}$ turns in Figure 1. In long dialogues, this shortcoming becomes more obvious and necessitates an approach for learning long-distance dependencies between dialogue turns.

To reason over dialogue context with long-distance dependencies, recent research in dialogues discovers graph-based structures at the turn level to predict the speaker's emotion (Ghosal et al., 2019) or generate sequential questions semi-autoregressively (Chai & Wan, 2020). Recently Zheng et al. (2019) incorporate graph neural models to connect the textual cues between all pairs of dialogue turns. These methods, however, involve a fixed graphical structure of dialogue turns, in which only a small number of nodes contains lexical overlap with the question of the current turn, e.g. the $1^{st}$, $3^{rd}$, and $5^{th}$ turns in Figure 1. These methods also fail to factor in the temporality of dialogue turns as the graph structures do not guarantee the sequential ordering among turns. In this paper, we propose a novel framework of Reasoning Paths in Dialogue Context (PDC). PDC model learns a reasoning path that traverses through dialogue turns to propagate contextual cues that are densely related to the semantics of the current questions. Our approach balances between a sequential and graphical process to exploit dialogue information.

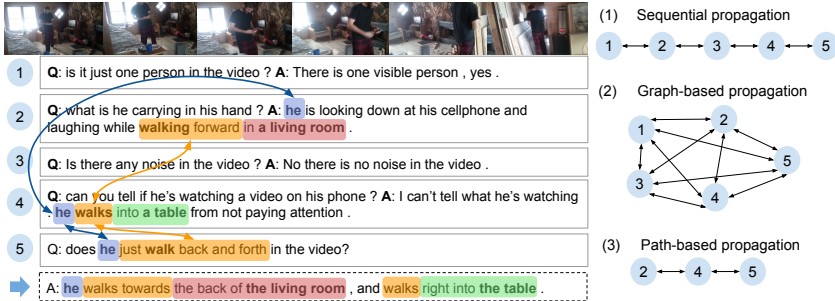

Figure 1: Sequential reasoning approaches fail to detect long-distance dependencies between the current turn and the $2^{nd}$ turn. Graph-based reasoning approaches signals from all turns are directly forwarded to the current turn but the $1^{st}$ and $3^{rd}$ contain little lexical overlap to the current question.

Our work is related to the long-studied research domain of discourse structures, e.g. (Barzilay & Lapata, 2008; Feng & Hirst, 2011; Tan et al., 2016; Habernal & Gurevych, 2017). A form of discourse structure is argument structures, including premises and claims and their relations. Argument structures have been studied to assess different characteristics in text, such as coherence, persuasiveness, and susceptibility to attack. However, most efforts are designed for discourse study in monologues and much less attention is directed towards conversational data. In this work, we investigate a form of discourse structure through semantic graphs built upon the overlap of component representations among dialogue turns. We further enhance the models with a reasoning path learning model to learn the best information path for the next utterance generation.

To learn a reasoning path, we incorporate our method with *bridge entities*, a concept often seen in reading comprehension research, and earlier used in entity-based discourse analysis (Barzilay & Lapata, 2008). In reading comprehension problems, bridge entities denote entities that are common between two knowledge bases e.g. Wikipedia paragraphs in HotpotQA (Yang et al., 2018b). In discourse analysis, entities and their locations in text are used to learn linguistic patterns that indicate certain qualities of a document. In our method, we first reconstruct each dialogue turn (including question and answer) into a set of component *sub-nodes* (e.g. entities, action phrases) using common syntactical dependency parsers. Each result dialogue turn contains sub-nodes that can be used as bridge entities. Our reasoning path learning approach contains 2 phases: (1) first, at each dialogue turn, a graph network is constructed at the turn level. Any two turns are connected if they have an overlapping sub-node or if two of their sub-nodes are semantically similar. (2) secondly, a path generator is trained to predict a path from the current dialogue turn to past dialogue turns that provide additional and relevant cues to answer the current question. The predicted path is used as a skeleton layout to propagate visual features through each step of the path.

Specifically, in PDC, we adopt non-parameterized approaches (e.g. cosine similarity) to construct the edges in graph networks and each sub-node is represented by pre-trained word embedding vectors. Our path generator is a transformer decoder that regressively generates the next turn index conditioned on the previously generated turn sequence. Our reasoning model is a combination of a vanilla graph convolutional network (Kipf & Welling, 2017) and transformer encoder (Vaswani et al., 2017). In each traversing step, we retrieve visual features conditioned by the corresponding dialogue turn and propagate the features to the next step. Finally, the propagated multimodal features are used as input to a transformer decoder to predict the answer.

Our experimental results show that our method can improve the results on the Audio-Visual Scene-Aware Dialogues (AVSD) generation settings (Alamri et al., 2019), outperform previous state-of-the-art methods. We evaluate our approach through comprehensive ablation analysis and qualitative study. PDC model also provides additional insights on how the inherent contextual cues in dialogue context are learned in neural networks in the form of a reasoning path.

## 2 RELATED WORK

**Discourses in monologues**. Related to our work is the research of discourse structures. A long-studied line of research in this domain focuses on argument mining to identify the structure of argument, claims and premises, and relations between them (Feng & Hirst, 2011; Stab & Gurevych, 2014; Peldszus & Stede, 2015; Persing & Ng, 2016; Habernal & Gurevych, 2017). More recently,

Ghosh et al. (2016); Duthie & Budzynska (2018); Jiang et al. (2019) propose to learn argument structures in student essays and official debates. In earlier approaches, Barzilay & Lapata (2008); Lin et al. (2011); Feng et al. (2014) study discourses to derive coherence assessment methods through entity-based representations of text. These approaches are proposed from linguistic theories surrounding entity patterns in discourses, i.e. how they are introduced and discussed (Grosz et al., 1995). Guinaudeau & Strube (2013); Putra & Tokunaga (2017) extend prior work with graphical structures in which sentence similarity is calculated based on semantic vectors representing those sentences. These lines of research show that studying discourse structures is useful in many tasks, such as document ranking and discrimination. However, most of these approaches are designed for monologues rather than dialogues.

**Discourses in dialogues**. More related to our problem setting is discourse research on text in a multi-turn setting. Murakami & Raymond (2010); Boltužić & Šnajder (2014); Swanson et al. (2015); Tan et al. (2016); Niculae et al. (2017); Morio & Fujita (2018); Chakrabarty et al. (2019) introduce new corpus and different methods to mine arguments in online discussion forums. Their models are trained to extract claims and premises in each user post and identify their relations between argument components in each pair of user posts. More recently, Li et al. (2020a); Jo et al. (2020) extend argument mining in online threads to identify attackability and persuasiveness in online posts.

In this work, we address the problem of video-grounded dialogue, in which dialogue turns are often semantically connected by a common grounding information source, a video. In this task, a discourse-based approach enables dialogue models to learn to anticipate the upcoming textual information in future dialogue turns. However, directly applying prior work on discourse or argument structures into video-grounded dialogues is not straightforward due to the inherent difference between online discussion posts and video-grounded dialogues. In video-grounded dialogues, the language is often closer to spoken language and there are fewer clear argument structures to be learned. Moreover, the presence of video necessitates the interaction between multiple modalities, text and vision. Incorporating traditional discourse structures to model cross-modality interaction is not straightforward. In this work, we propose to model dialogue context by using compositional graphical structures and constructing information traversal paths through dialogue turns.

**Graph-based dialogue models**. Related to our work is research study that investigates different types of graph structures in dialogue. Hu et al. (2019); Shi & Huang (2019); Zhu et al. (2020) address the "reply_to" relationship among multi-party dialogues through graph networks that incorporate conversational flows in comment threads on social networks, e.g. Reddit and Ubuntu IRC, and online games. Zheng et al. (2019) propose a fully connected graph structure at the turn level for visual dialogues. Concurrently, Ghosal et al. (2019) also propose a fully connected graph structure with heterogeneous edges to detect the emotion of participating speakers. All of these methods discover graph structures connecting pairs of dialogue turns of little lexical overlap, resulting in sub-optimal feature propagation. This drawback becomes more significant in question answering problems in multi-turn settings. Our approach constructs graph networks based on compositional similarities.

**Reasoning path learning**. Our method is also motivated by the recent research of machine reading comprehension, e.g. WikiHop (Welbl et al., 2018) and HotpotQA (Yang et al., 2018a). De Cao et al. (2019); Qiu et al. (2019) construct graph networks of supporting documents with entity nodes that are connected based on different kinds of relationships. Tu et al. (2019); Tang et al. (2020) enhance these methods with additional edges connecting output candidates and documents. Extended from these methods are path-based approaches that learn to predict a reasoning path through supporting documents. Kundu et al. (2019); Asai et al. (2020) score and rank path candidates that connect entities in question to the target answer. A common strategy among these methods is the use of *bridge entities*. However, unlike reading comprehension, dialogues are normally not entity-centric and it is not trivial to directly adopt bridge entities into dialogue context.

**Cross-modality feature learning**. Our work is related to study that integrates visual and linguistic information representation. A line of research in this domain is the problem of visual QA, e.g. (Minh Le et al., 2020; Gao et al., 2019). Closer to our method are methods that adopt compositionality in textual features. Specifically, Socher et al. (2014) introduce image and language representation learning by detecting the component lexical parts in sentences and combining them with image features. The main difference between these approaches and our work is the study of cross-modalities in a multi-turn setting. Our approach directly tackles the embedded sequential order in dialogue utterances and examines how cross-modality features are passed from turn to turn.

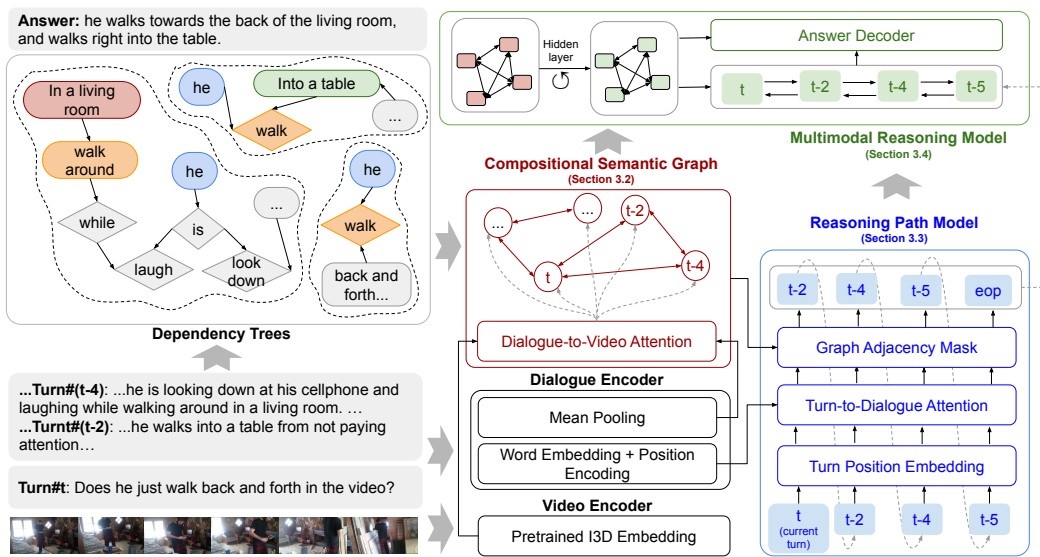

Figure 2: An overview of our PDC method.

# 3 METHOD

To describe our PDC model, we introduce a new graph-based method (Section 3.2) that constructs a graph structure to connect turn-level representations in dialogue context based on their compositional semantics. The compositional semantics consists of sub-nodes detected through syntactical dependency parsing methods. We enhance our approach with a path-based propagation method (Section 3.3) to narrow down the contextual information that facilitates question answering of the current turn. Our approach integrates a strong strategy to model dialogue flows in the form of graphical and path-based information such that contextual linguistic information is exploited to propagate relevant visual features (Section 3.4). Figure 2 demonstrates an overview of our method.

## 3.1 PROBLEM DEFINITION

The inputs to a question answering problem in a multi-turn setting consist of a dialogue $\mathcal{D}$ and the visual input of a video $\mathcal{I}$. Each dialogue contains a sequence of dialogue turns, each of which is a pair of question $\mathcal{Q}$ and answer $\mathcal{A}$. At each dialogue turn $t$, we denote the dialogue context $\mathcal{C}_t$ as all previous dialogue turns $\mathcal{C}_t = \{(\mathcal{Q}_i, \mathcal{A}_i)\}|_{i=1}^{i=t-1}$. Since it is positioned in a dialogue, the question of turn $t$ $\mathcal{Q}_t$ might be dependent on a subset of the dialogue context $\mathcal{C}_t$. The output is the answer of the current turn $\hat{\mathcal{A}}_t$. Each textual component, i.e. $\mathcal{Q}$ and $\mathcal{A}$, is represented as a sequence of token or word indices $\{w_m\}|_{m=1}^{m=L} \in |\mathbb{V}|$, where $L$ is the sequence length and $\mathbb{V}$ is the vocabulary set. The objective of the task is the generation objective that output answers of the current turn:

$$\hat{\mathcal{A}}_t = \arg\max_{\mathcal{A}_t} P(\mathcal{A}_t|\mathcal{I}, \mathcal{C}_t, \mathcal{Q}_t; \boldsymbol{\theta}) = \arg\max_{\mathcal{A}_t} \prod_{m=1}^{L_{\mathcal{A}}} P_m(w_m|\mathcal{A}_{t,1:m-1}, \mathcal{I}, \mathcal{C}_t, \mathcal{Q}_t; \boldsymbol{\theta}) \quad (1)$$

## 3.2 COMPOSITIONAL SEMANTIC GRAPH OF DIALOGUE CONTEXT

The semantic relations between dialogue turns are decomposed to semantic relations between sub-nodes that constitute each turn. These composition relations serve as strong clues to determine how a dialogue turn is related to another. We first employ a co-reference resolution system, e.g. (Clark & Manning, 2016), to replace pronouns with the original entities. We then explore using the Stanford parser system[1] to discover sub-nodes. The parser decomposes each sentence into grammatical components, where a word and its modifier are connected in a tree structure. For each dialogue turn, we concatenate the question and answer of that turn as input to the parser. The output dependency tree is pruned to remove unimportant constituents and merge adjacent nodes to form a semantic unit.

---

[1]v3.9.2 retrieved at `https://nlp.stanford.edu/software/lex-parser.shtml`

A graph structure $\mathcal{G}$ is then constructed. Any two turns are connected if one of their corresponding sub-nodes are semantically similar. To calculate the similarity score, we obtain their pre-trained word2vec embeddings[2] and compute the cosine similarity score. Algorithm 1 provides the details of the procedure to automatically construct a semantic graph. Note that our approach can also be applied with other co-reference resolution systems, parser, or pre-trained embeddings. Unlike graph structures in machine reading comprehension such as Wikipedia graph, the semantic graph $\mathcal{G}$ is not fixed throughout the sample population but is constructed for each dialogue and at each turn.

---

**Algorithm 1:** Compositional semantic graph of dialogue context

---

**Data:** Dialogue context $\mathcal{C}_t$, question of the current turn $\mathcal{Q}_t$
**Result:** Semantic graph $\mathcal{G} = (\mathcal{V}, \mathcal{E})$

1 **begin**
2     $\mathcal{T} \longleftarrow \emptyset; \mathcal{G} = \{\mathcal{V}, \mathcal{E}\}; \mathcal{E} \longleftarrow \emptyset; \mathcal{V} \longleftarrow \emptyset; \mathcal{S} \longleftarrow \emptyset;$
3     $\mathcal{H} \longleftarrow \text{Coreference\_Resolution}([\mathcal{C}_t; \mathcal{Q}_t]);$
4     **for each dialogue turn** $h \in \mathcal{H}$ **do**
5        $T_h \longleftarrow \text{Merge\_Nodes}(\text{Prune\_Tree}(\text{Dependency\_Parse}(h))); \mathcal{T} \longleftarrow \mathcal{T} \cup \{T_h\};$
6        $\mathcal{V} \longleftarrow \mathcal{V} \cup \{h\}; \mathcal{E} \longleftarrow \mathcal{E} \cup \{\langle \text{Turn\_Position}(h), \text{Turn\_Position}(h) \rangle\}$
7     **for each dependency tree** $T = (V_T, E_T) \in \mathcal{T}$ **do** $\mathcal{S} \longleftarrow \mathcal{S} \cup \{V_T\}$
8     **for each sub-node** $s_i \in \mathcal{S}$ **do**
9        **for each sub-node** $s_j \in \mathcal{S}$ **do**
10           **if not** $\text{In\_Same\_Turn}(s_i, s_j)$ **and** $\text{Is\_Similar}(s_i, s_j)$ **then**
11              $\mathcal{E} \longleftarrow \mathcal{E} \cup \{\langle \text{Get\_Dial\_Turn}(s_i), \text{Get\_Dial\_Turn}(s_j) \rangle\}$
12              $\mathcal{E} \longleftarrow \mathcal{E} \cup \{\langle \text{Get\_Dial\_Turn}(s_j), \text{Get\_Dial\_Turn}(s_i) \rangle\}$

13     **return** $\mathcal{G}$

---

### 3.3 LEARNING TO GENERATE REASONING PATHS

Our proposed compositional approach to construct a semantic graph in dialogue context ensures lexical overlaps with the question, but the graph structure does not guarantee the temporal order of dialogue turns. To ensure this sequential information is maintained, we train a generator to predict reasoning paths that traverse through current dialogue turn to past dialogue turns.

We use a Transformer decoder to model the reasoning paths from the current turn $t$. The first position of the path, $z_0$ is initialized with the turn-level position embedding of $t$. The next turn index is generated auto-regressively by conditioning on the previously generated path sequence:

$$z_0 = \text{Embed}(t) \in \mathbb{R}^d \tag{2}$$

$$Z_{0:m-1} = \text{Embed}([t; \hat{r}_1, ..., \hat{r}_{m-1}]) \tag{3}$$

where $\hat{r}_i$ denotes a predicted dialogue turn index. The dialogue context and question of the current turn are represented by embedding vectors of their component tokens. Following Vaswani et al. (2017), their representations are enhanced with the sine-cosine positional encoding PosEncode.

$$Q_t = \text{Embed}(\mathcal{Q}_t) + \text{PosEncode}(\mathcal{Q}_t) \in \mathbb{R}^{L_{\mathcal{Q}_t} \times d} \tag{4}$$

$$C_t = \text{Embed}(\mathcal{C}_t) + \text{PosEncode}(\mathcal{C}_t) \in \mathbb{R}^{L_{\mathcal{C}_t} \times d} \tag{5}$$

Note that the dialogue context representation $C_t$ is the embedding of dialogue turns up to the last turn $t - 1$, excluding answer embedding of the current turn $A_t$.

We denote a Transformer attention block as $\text{Transformer}(\text{query}, \text{key}, \text{value})$. The path generator incorporates contextual information through attention layers on dialogue context and question.

$$D_{\text{path}}^{(1)} = \text{Transfromer}(Z_{0:m-1}, Z_{0:m-1}, Z_{0:m-1}) \in \mathbb{R}^{m \times d} \tag{6}$$

$$D_{\text{path}}^{(2)} = \text{Transfromer}(D_{\text{path}}^{(1)}, Q_t, Q_t) \in \mathbb{R}^{m \times d} \tag{7}$$

$$D_{\text{path}}^{(3)} = \text{Transfromer}(D_{\text{path}}^{(2)}, C_t, C_t) \in \mathbb{R}^{m \times d} \tag{8}$$

---

[2] https://code.google.com/archive/p/word2vec/

At the $m$-th decoding step ($m \geq 1$), our model selects the next dialogue turn among the set of dialogue turns that are adjacent to one at $(m-1)$-th decoding step in the semantic graph. This is enforced through masking the softmax output scores in which non-adjacent turn indices are assigned to a very low scalar $s_{\text{masked}}$. We denote the adjacency matrix of semantic graph $\mathcal{G} = (\mathcal{V}, \mathcal{E})$ as a square matrix $A$ of size $|\mathcal{V}| \times |\mathcal{V}|$ where $A_{i,j} = 1$ if $\langle i, j \rangle \in \mathcal{E}$ and $A_{i,i} = 1 \forall i = 1, ..., |\mathcal{V}|$. The probability of decoded turns at the $m$-th decoding step is:

$$P_m = \text{softmax}(D^{(3)}_{\text{path},m} W_{\text{path}}) \in \mathbb{R}^{|V|}, \quad P_{m,i} = s_{\text{masked}} \forall i | A_{\hat{r}_{m-1}, i} = 0 \tag{9}$$

where $W_{\text{path}} \in \mathbb{R}^{d \times |V|}$. The decoding process is terminated when the next decoded token is an *[EOP]* (end-of-path) token. During inference time, we adopt a greedy decoding approach. Due to the small size of $\mathcal{V}$, we found that a greedy approach can perform as well as beam search methods. The computational cost of generating reasoning paths in dialogue context is, thus, only dependent on the average path length, which is bounded by the maximum number of dialogue turns.

**Data Augmentation**. We train our path generator in a supervision manner. At each dialogue turn $t$ with a semantic graph $\mathcal{G}$, we use a graph traversal method, e.g. BFS, to find all paths that start from the current turn to any past turn. We maintain the ground-truth paths with dialogue temporal order by keeping the dialogue turn index in path position $m$ lower than the turn index in path position $m - 1$. We also narrow down ground-truth paths based on their total lexical overlaps with the expected output answers. Using the dialogue in Figure 1 as an example, using BFS results in three potential path candidates: $5 \rightarrow 4$, $5 \rightarrow 2$, and $5 \rightarrow 4 \rightarrow 2$. We select $5 \rightarrow 4 \rightarrow 2$ as the ground-truth path because it can cover the most sub-nodes in the expected answers. If two paths have the same number of lexical overlaps, we select one with a shorter length. If two paths are equivalent, we randomly sample one path following uniform distribution at each training step. Ground-truth reasoning paths are added with *[EOP]* token at the final position for termination condition. The objective to train the path generator is the generation objective of reasoning path at each dialogue turn:

$$\hat{\mathcal{R}}_t = \arg\max_{\mathcal{R}_t} P(\mathcal{R}_t | \mathcal{C}_t, \mathcal{Q}_t; \phi) = \arg\max_{\mathcal{R}_t} \prod_{m=1}^{L_{\text{path}}} P_m(r_m | \mathcal{R}_{t,1:m-1}, \mathcal{C}_t, \mathcal{Q}_t; \phi) \tag{10}$$

### 3.4 Multimodal Reasoning from Reasoning Paths

The graph structure $\mathcal{G}$ and generated path $\hat{\mathcal{R}}_t$ are used as layout to propagate features of both textual and visual inputs. For each dialogue turn from $\mathcal{V}$, we obtain the corresponding embeddings and apply mean pooling to get a vector representation. We denote the turn-level representations of $\mathcal{V}$ as $V \in \mathbb{R}^{d \times |V|}$. We use attention to retrieve the turn-dependent visual features from visual input.

$$M = \text{Transformer}(V, I, I) \in \mathbb{R}^{d \times |V|} \tag{11}$$

where $I$ is a two-dimensional feature representation of visual input $\mathcal{I}$. We define a new multi-modal graph based on semantic graph $\mathcal{G}$: $G_{\text{mm}} = (\mathcal{V}_{\text{mm}}, \mathcal{E}_{\text{mm}})$ where $\mathcal{V}_{\text{mm}} = M$ and edges $\langle m_i, m_j \rangle \in \mathcal{E}_{\text{mm}} \forall i, j | \langle i, j \rangle \in \mathcal{E}$. We employ a vanilla graph convolution network (Kipf & Welling, 2017) to update turn-level multimodal representations through message passing along all edges.

$$e_k = \frac{1}{|\Omega_k|} \sum_{m_j \in \Omega_k} f(m_k, m_j), \quad e = \frac{1}{|V|} \sum_k e_k, \quad \widetilde{m}_k = g(m_k, e_k, e) \tag{12}$$

where $\Omega_k$ is the set of adjacent nodes of $m_k$ and $f(.)$ and $g(.)$ are non-linear layers, e.g. MLP and their inputs are just simply concatenated. To propagate features along a reasoning path $\hat{\mathcal{R}}_t$, we utilize the updated turn-level multimodal representations $\widetilde{M} \in |V|$ and traverse the path sequentially through the representation of the corresponding turn index $r_m$ in each traversing step. Specifically, We obtain $G = \{\widetilde{m}_{\hat{r}_0}, \widetilde{m}_{\hat{r}_1}...\} \in \mathbb{R}^{L_{\text{path}} \times d}$. The traversing process can be done through a recurrent network or a transformer encoder.

$$\widetilde{G} = \text{Transformer}(G, G, G) \in \mathbb{R}^{L_{\text{path}} \times d} \tag{13}$$

To incorporate propagated features into the target response, we adopt a state-of-the-art decoder model from (Le et al., 2019) that exploits multimodal attention over contextual features. Specifically, We integrate both $\widetilde{M}$ and $\widetilde{G}$ at each response decoding step through two separate attention layers.

| Models | B-1 | B-2 | B-3 | B-4 | M | R | C |
|---|---|---|---|---|---|---|---|
| Baseline (Hori et al., 2019) | 0.621 | 0.480 | 0.379 | 0.305 | 0.217 | 0.481 | 0.733 |
| TopicEmb (Kumar et al., 2019)‡ | 0.632 | 0.499 | 0.402 | 0.329 | 0.223 | 0.488 | 0.762 |
| FGA (Schwartz et al., 2019b) | - | - | - | - | - | - | 0.806 |
| JMAN (Chu et al., 2020) | 0.667 | 0.521 | 0.413 | 0.334 | 0.239 | 0.533 | 0.941 |
| FA+HRED (Nguyen et al., 2019)‡ | 0.695 | 0.533 | 0.444 | 0.360 | 0.249 | 0.544 | 0.997 |
| MTN (Le et al., 2019) | 0.715 | 0.581 | 0.476 | 0.392 | 0.269 | 0.559 | 1.066 |
| VideoSum (Sanabria et al., 2019)† | 0.718 | 0.584 | 0.478 | 0.394 | 0.267 | 0.563 | 1.094 |
| MSTN (Lee et al., 2020)‡ | - | - | - | 0.377 | 0.275 | 0.566 | 1.115 |
| Student-Teacher (Hori et al., 2019)‡ | 0.727 | 0.593 | 0.488 | 0.405 | 0.273 | 0.566 | 1.118 |
| **PDC (Ours)** | **0.747** | **0.616** | **0.512** | **0.429** | **0.282** | **0.579** | **1.194** |
| VideoSum (Sanabria et al., 2019)†§ | 0.723 | 0.586 | 0.476 | 0.387 | 0.266 | 0.564 | 1.087 |
| VGD-GPT2 (Le & Hoi, 2020)†§ | 0.749 | 0.620 | 0.520 | 0.436 | 0.282 | 0.582 | 1.194 |
| RLM (Li et al., 2020b)‡§ | 0.765 | 0.643 | **0.543** | **0.459** | **0.294** | **0.606** | **1.308** |
| **PDC (Ours) + GPT2**§ | **0.770** | **0.653** | 0.539 | 0.449 | 0.292 | **0.606** | 1.295 |

Table 1: **AVSD@DSTC7 test results:** † uses visual features other than I3D, e.g. ResNeXt, scene graphs. ‡ incorporates additional video background audio inputs. § indicates finetuning methods on additional data or pre-trained language models. Metric notations: B-n: BLEU-n, M: METEOR, R: ROUGE-L, C: CIDEr.

Besides, we also experiment with integrating propagated features with decoder as Transformer language models. Transformer language models have shown impressive performance recently in generation tasks by transferring language representations pretrained in massive data (Radford et al., 2019). To integrate, we simply concatenate $\widetilde{M}$ and $\widetilde{G}$ to the input sequence embeddings as input to language models, similar as (Le & Hoi, 2020; Li et al., 2020b).

**Optimization**. The multimodal reasoning model is learned jointly with other model components. All model parameters are optimized through the objectives from both Equation 1 and 10. We use the standard cross-entropy loss which calculates the logarithm of each softmax score at each decoding position of $\hat{\mathcal{A}}_t$ and $\hat{\mathcal{R}}_t$.

## 4 EXPERIMENTS

**Dataset**. We use the Audio-Visual Sene-Aware Dialogue (AVSD) benchmark developed by Alamri et al. (2019). The benchmark focuses on dialogues grounded on videos from the Charades dataset (Sigurdsson et al., 2016). Each dialogue can have up to 10 dialogue turns, which makes it an appropriate choice to evaluate our approach of reasoning paths over dialogue context. We used the standard visual features I3D to represent the video input. We experimented with the test splits used in the $7^{th}$ Dialogue System Technology Challenge (DSTC7) (Yoshino et al., 2019) and DSTC8 (Kim et al., 2019). Please see the Appendix A for our experimental setups.

| | Train | Val | Test@DSTC7 | Test@DSTC8 |
|---|---|---|---|---|
| **#Dialogs** | 7,659 | 1,787 | 1,710 | 1,710 |
| **#Questions/Answers** | 153,180 | 35,740 | 13,490 | 18,810 |
| **#Words** | 1,450,754 | 339,006 | 110,252 | 162,226 |

Table 2: Dataset Summary of the AVSD benchmark with both test splits @DSTC7 and @DSTC8.

**Overall Results.** The dialogues in the AVSD benchmark focuses on question answering over multiple turns and entail less semantic variance than open-domain dialogues. Therefore, we report the objective scores, including BLEU (Papineni et al., 2002), METEOR (Banerjee & Lavie, 2005), ROUGE-L (Lin, 2004), and CIDEr (Vedantam et al., 2015), which are found to have strong correlation with human subjective scores (Alamri et al., 2019). In Table 1 and 3, we present the test results of our models in comparison with previous models in DSTC7 and DSTC8 respectively. In both test splits, our models achieve very strong performance against models without using pre-trained language models. Comparing with models using pre-trained models and additional fine-tuning, our models achieve competitive performances in both test splits. The performance gain of our models when using GPT2 indicates current model sensitivity to language modelling as a generator. A unique benefit of our

| Models | B-1 | B-2 | B-3 | B-4 | M | R | C |
|---|---|---|---|---|---|---|---|
| Baseline (Hori et al., 2019) | 0.614 | 0.467 | 0.365 | 0.289 | 0.210 | 0.480 | 0.651 |
| DMN (Xie & Iacobacci, 2020)[‡] | - | - | - | 0.296 | 0.214 | 0.496 | 0.761 |
| Simple (Schwartz et al., 2019a) | - | - | - | 0.311 | 0.224 | 0.502 | 0.766 |
| JMAN (Chu et al., 2020) | 0.645 | 0.504 | 0.402 | 0.324 | 0.232 | 0.521 | 0.875 |
| STSGR (Geng et al., 2020)[†] | - | - | - | 0.357 | 0.267 | 0.553 | 1.004 |
| MSTN (Lee et al., 2020)[‡] | - | - | - | 0.385 | **0.270** | 0.564 | 1.073 |
| **PDC(Ours)** | **0.723** | **0.595** | **0.493** | **0.410** | **0.270** | **0.570** | **1.105** |
| RLM (Li et al., 2020b) [‡§] | 0.746 | 0.626 | **0.528** | **0.445** | **0.286** | **0.598** | **1.240** |
| **PDC (Ours) + GPT2**[§] | **0.749** | **0.629** | **0.528** | 0.439 | 0.285 | 0.592 | 1.201 |

Table 3: **AVSD@DSTC8 test results:** † uses visual features other than I3D, e.g. ResNeXt, scene graphs. ‡ incorporates additional video background audio inputs. § indicates finetuning methods on additional data or pre-trained language models. Metric notations: B-n: BLEU-n, M: METEOR, R: ROUGE-L, C: CIDEr.

| Semantics | Direction | GraphProp | PathProp | B-1 | B-2 | B-3 | B-4 | M | R | C |
|---|---|---|---|---|---|---|---|---|---|---|
| Comp. | BiDirect | ✓ | ✓ | 0.747 | **0.616** | **0.512** | **0.429** | **0.282** | 0.579 | **1.194** |
| Comp. | BiDirect | ✓ | | 0.746 | 0.611 | 0.503 | 0.418 | 0.281 | **0.580** | 1.179 |
| Comp. | TODirect | ✓ | | 0.748 | 0.613 | 0.505 | 0.420 | 0.279 | 0.579 | 1.181 |
| Comp. | BiDirect | | ✓ | 0.745 | 0.611 | 0.504 | 0.419 | **0.282** | 0.577 | 1.172 |
| Global | BiDirect | ✓ | ✓ | 0.743 | 0.609 | 0.504 | 0.421 | 0.279 | 0.579 | 1.178 |
| Global | BiDirect | ✓ | | 0.744 | 0.610 | 0.502 | 0.416 | 0.280 | 0.577 | 1.169 |
| Global | TODirect | ✓ | | 0.743 | 0.609 | 0.501 | 0.416 | 0.279 | 0.579 | 1.161 |
| Global | BiDirect | | ✓ | **0.749** | 0.613 | 0.505 | 0.421 | 0.279 | 0.578 | 1.172 |
| Fully | BiDirect | ✓ | | 0.745 | 0.607 | 0.500 | 0.414 | 0.277 | 0.576 | 1.169 |
| Fully | TODirect | ✓ | | 0.743 | 0.605 | 0.497 | 0.411 | 0.277 | 0.573 | 1.163 |

Table 4: **Ablation of AVSD@DSTC7 test results:** We experiment with graphs that are compositional semantics, global semantics, and fully-connected with bidirectional or temporally ordered edges, and with graph-based or path-based feature propagation. Metric notations: B-n: BLEU-n, M: METEOR, R: ROUGE-L, C: CIDEr.

models from prior approaches is the insights of how the models exploit information from dialogue turns in the form of reasoning paths (Please see example outputs in Figure 3).

**Ablation Analysis**. In Table 4 we report the results of path learning in a *global* semantic graph. In these graphs, we do not decompose each dialogue turn into component sub-nodes (line 5 in Algorithm 1) but directly compute the similarity score based on the whole sentence embedding. In this case, to train the path generator, we obtain the ground-truth path by using BFS to traverse to the node with the most sentence-level similarity score to the expected answer. We observe that: (1) models that learn paths based on component lexical overlaps results in better performance than paths based on global lexical overlaps in most of the objective metrics. (2) Propagation by reasoning path alone without using GCN does not result in better performance. This can be explained as the information in each traversal step is not independent but still contains semantic dependencies to other turns. It is different from standard reading comprehension problems where each knowledge base is independent and it is not required to propagate features through a graph structure to obtain contextual updates. Please see the Appendix B for additional analysis of Table 4.

**Impacts of Reasoning Path Learning**. We compare models that can learn reasoning paths against those that use a fixed propagation path through the past dialogue turns. From Table 5, we observe that: (1) learning dynamic instance-based reasoning paths outperforms all models that propagate through a default path. This is achieved by using the reasoning path as a skeleton for feature propagation as well as adopting the joint training strategy. We can consider dynamically learned paths as an ideal traversal path to propagate visual cues among all possible paths within the semantic graph of the dialogue context. (2) our path generator can generate reasoning paths well and the model with learned paths can perform as well as one using the oracle paths. (3) due to the short length of reasoning paths (limited by the maximum dialogue length), either beam search or greedy decoding approach is good enough to generate paths. The greedy approach has the advantage of much lower computational cost.

**Qualitative Analysis.** In Figure 3, we demonstrate some examples of our predicted responses and the corresponding reasoning paths. Specifically, we showcase samples in which the reasoning paths are 2-hops (Example A and B) and 3-hops (Example C and D), and the distance in each hop can be over one dialogue turn (Example B and D) or more (Example A and C). The example reasoning paths

| Reasoning Path | B-1 | B-2 | B-3 | B-4 | M | R | C |
|---|---|---|---|---|---|---|---|
| Learned Path (beam search) | 0.747 | 0.616 | **0.512** | 0.429 | **0.282** | 0.579 | 1.194 |
| Learned Path (greedy) | 0.747 | 0.616 | **0.512** | **0.430** | **0.282** | **0.580** | **1.195** |
| Oracle Path | 0.748 | **0.617** | **0.512** | **0.430** | **0.282** | **0.580** | **1.195** |
| Random Path | 0.552 | 0.437 | 0.345 | 0.274 | 0.194 | 0.420 | 0.684 |
| Path through last 10 turns | 0.744 | 0.607 | 0.500 | 0.415 | 0.278 | 0.576 | 1.166 |
| Path through last 9 turns | 0.743 | 0.607 | 0.500 | 0.416 | 0.277 | 0.574 | 1.161 |
| Path through last 8 turns | 0.749 | 0.615 | 0.509 | 0.423 | 0.277 | 0.578 | 1.168 |
| Path through last 7 turns | **0.754** | 0.618 | 0.510 | 0.422 | **0.282** | 0.579 | 1.170 |
| Path through last 6 turns | 0.746 | 0.608 | 0.498 | 0.412 | 0.278 | 0.575 | 1.150 |
| Path through last 5 turns | 0.744 | 0.607 | 0.500 | 0.415 | 0.278 | 0.576 | 1.169 |
| Path through last 4 turns | 0.745 | 0.610 | 0.502 | 0.417 | 0.278 | 0.576 | 1.165 |
| Path through last 3 turns | 0.744 | 0.607 | 0.500 | 0.414 | 0.278 | 0.576 | 1.163 |
| Path through last 2 turns | 0.748 | 0.615 | 0.508 | 0.423 | 0.278 | 0.579 | 1.171 |
| Path through last 1 turns | 0.740 | 0.603 | 0.494 | 0.408 | 0.276 | 0.575 | 1.149 |

Table 5: **Comparison of AVSD@DSTC7 test results between learned paths and paths as sequences of the last $n$ turns:** We experiment with paths predicted by our path generator and paths as a sequence of the last $n$ turns, i.e. $\{t, ..., \max(0, t - n)\}$. Metric notations: B-n: BLEU-n, M: METEOR, R: ROUGE-L, C: CIDEr.

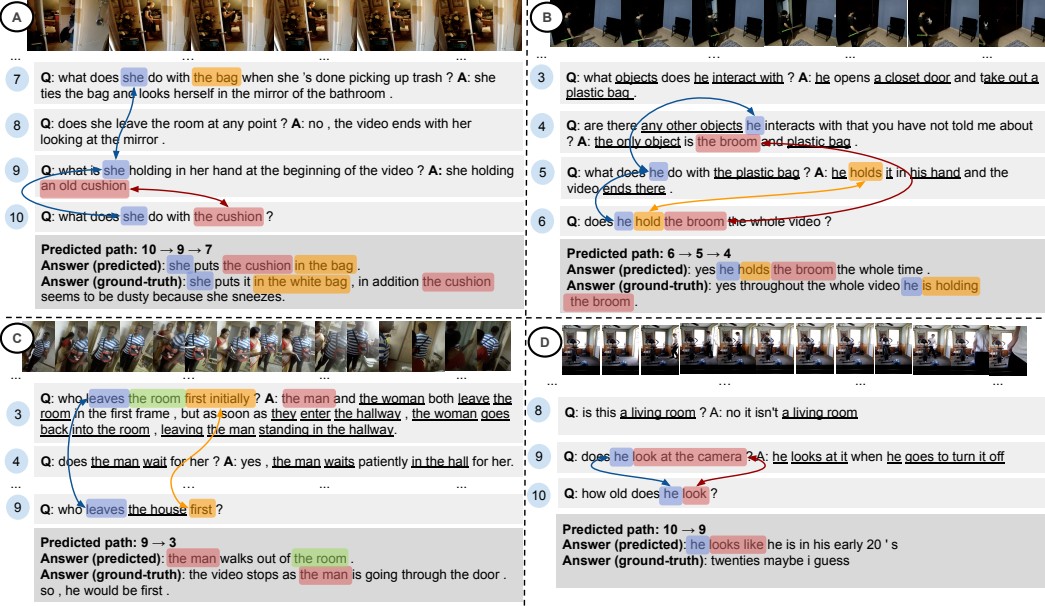

Figure 3: **Example outputs of reasoning paths and dialogue responses.** We demonstrate 4 cases of reasoning paths with 2 to 3 hops and with varied distances between two ends of the reasoning path.

show to be able to connect a sequence of dialogue turns that are most relevant to questions of the current turn. For instance, in Example A, the reasoning path can connect the $7^{th}$ and $9^{th}$ turn to the current turn as they contain lexical overlaps, i.e. "the bag", and "the cushion". The path skips the $8^{th}$ turn which is not relevant to the current question. Likewise, in Example C, the path skips the $4 - 8^{th}$ turns. All examples show that dialogue context can be used to extract additional visual clues relevant to the current turn. Information from dialogues, thus, deserves more attention than just being used as a background text input. Please see the Appendix C for additional analysis.

## 5  CONCLUSION

We proposed PDC, a novel approach to learning a reasoning path over dialogue turns for video-grounded dialogues. Our approach exploits the compositional semantics in each dialogue turn to construct a semantic graph, which is then used to derive an optimal path for feature propagation. Our experiments demonstrate that our model can learn to retrieve paths that are most relevant to the current question. We hope our approach can motivate further study to investigate reasoning over multiple turns, especially in complex settings with interconnected dialogue flows (Sun et al., 2019).

ACKNOWLEDGMENTS

We thank all reviewers for their insightful feedback on the manuscript of this paper. The first author of this paper is supported by the Agency for Science, Technology and Research (A*STAR) Computing and Information Science scholarship.

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

## A  EXPERIMENTAL SETUP

We experiment with the Adam optimizer (Kingma & Ba, 2015). The models are trained with a warm-up learning rate period of 5 epochs before the learning rate decays and the training finishes up to 50 epochs. The best model is selected by the average loss in the validation set. All model parameters, except the decoder parameters when using pre-trained language models, are initialized with uniform distribution (Glorot & Bengio, 2010). The Transformer hyper-parameters are fine-tuned by validation results over $d = \{128, 256\}$, $h = \{1, 2, 4, 8, 16\}$, and a dropout rate from $0.1$ to $0.5$. Label smoothing (Szegedy et al., 2016) is applied on labels of $\hat{\mathcal{A}}_t$ (label smoothing does not help when optimizing over $\hat{\mathcal{R}}_t$ as the labels are limited by the maximum length of dialogues, i.e. 10 in AVSD).

## B  IMPACTS OF COMPOSITIONAL SEMANTIC GRAPH

We experiment with model variants based on different types of graph structures. Specifically, we compare our compositional semantic graph against a graph built upon the turn-level *global* semantics. In these graphs, we do not decompose each dialogue turn into component sub-nodes (line 5 in Algorithm 1) but directly compute the similarity score based on the whole sentence embedding. We also experiment with a fully connected graph structure. In each graph structure, we experiment with temporally ordered edges (TODirect). This is enforce by adding a check whether $\text{Get\_Dial\_Turn}(s_j) > \text{Get\_Dial\_Turn}(s_i)$ in line 11 and removing line 12 in Algorithm 1. From the results in Table 4, we observe that: (1) based on the CIDEr metric, the best performing graph structure is the compositional semantic graph while the global semantic graph and fully connected graph structure are almost equivalent. This is consistent with the previous insight in machine reading comprehension research that entity lexical overlaps between knowledge bases are often overlooked by global embeddings (Ding et al., 2019) and it is not reliable to construct a knowledge graph based on global representations alone. (2) regarding the direction of edges, bidirectional edges and temporally ordered edges perform similarly, indicating that processing dialogue turns following temporal orders provides enough information and backward processing is only supplementary.

## C  ADDITIONAL QUALITATIVE ANALYSIS

In Figure 4, we demonstrate examples outputs of reasoning paths and dialogue responses and have the following observations:

- For questions that do not involve actions and can be answered by a single frame, there is typically no reasoning path, i.e. the path only includes the current turn (Example A and B). These questions are usually simple and they are rarely involved in multiple dialogue turns.

- In many cases, the dialogue agent can predict an appropriate path but still not generate the correct answers (Example D and G). These paths are able to connect turns that are most relevant to the current turns but these past turns do not contain or contain very limited clues to the expected answers. For example, in Example F, the $2^{nd}$ and $4^{th}$ turn are linked by the lexical component for "the woman". However, they do not have useful information relevant to the current turn, i.e. her clothes.

- Finally, our approach shows that the current benchmark, AVSD, typically contains one-hop (Example C, D, E) to two-hop (Example F, G, H) reasoning paths over dialogue context. We hope future dialogue benchmarks will factor in the complexity of dialogue context in terms of reasoning hops to facilitate better research of intelligent dialogue systems.

**Discussion of failure cases**. From the above observations, we identify the following scenarios that our models are susceptible to and propose potential directions for improvement.

- **Long complex utterances**. One limitation of our methods is its dependence on syntactical parser methods to decompose a sentence into sub-nodes. In most dialogues, this problem is not too serious due to the short length of utterances, usually just a single sentence. However, in cases that the utterance contains multiple sentences/clauses or exhibits usage of spoken language with loose linguistic syntax, the parser may fail to decompose it properly. For

instance, in Example G in Figure 4, the ground-truth answer contains a causality-based clause ("because"), making it harder to identify sub-nodes such as "sneeze" or "dusty".

- **Contextualized semantic similarity**. Another area we can improve upon this method is to inject some forms of sentence-level contextual cues into each sub-node to improve their semantic representations. For instance, in a hypothetical dialogue that involves 2 question utterances such as the $2^{nd}$ turn in Example A and the $6^{th}$ turn in Example E in Figure 4, our method might not detect the connection between these two as they do not have overlap component sub-nodes. However, they are both related to the audio aspect of the video and a reasoning path between these two turns is appropriate.

## D    STATISTICS OF LOCAL VS. GLOBAL SEMANTIC GRAPHS

In Table 6, we report the statistics of graph structures constructed by local and global semantics in all data splits of the AVSD benchmark. We observe that constructing graphs with local semantics result in a lower number of instances with no reasoning paths than making graphs with global semantics. This is due to compositionality in our method, resulting in higher lexical overlap between dialogue turns. With our method, the number of sub-nodes per dialogue turn is more than 4 on average, making it easier to connect dialogue turns. This also leads to a larger and more diverse set of reasoning paths for supervision learning. In local semantic graphs, the average number of reasoning paths per dialogue turn is 2 to 3 on average, higher than this number in global semantic graphs. Although our method requires additional computational effort to constructing these graphs, it is scalable to the size of the dialogue, i.e. number of the dialogue turns. To efficiently construct these graphs in a dialogue, the semantic graph of a dialogue turn can be built on top of the semantic graph of the last turn. This is done by simply adding the new sub-nodes to the last turn's semantic graph and defining new edges adjacent to these sub-nodes only. In this way, the complexity of our graph construction method is linear to the number of dialogue turns.

|  | Local Semantic Graphs | | | Global Semantic Graphs | | |
|---|---|---|---|---|---|---|
|  | train | val | test | train | val | test |
| # sub-nodes in total | 341,186 | 88,617 | 31,936 | 76,590 | 17,870 | 6,745 |
| # sub-nodes per dialogue turn | 4.45 | 4.96 | 4.73 | 1.00 | 1.00 | 1.00 |
| # edges in total | 619,428 | 147,588 | 32,765 | 270,356 | 63,141 | 40,470 |
| # dialogue turns with no reasoning paths | 27,704 | 6,420 | 3,048 | 35,048 | 8,152 | 3,696 |
| max # reasoning paths per dialogue turn | 35 | 33 | 24 | 10 | 9 | 5 |
| avg. # reasoning paths per dialogue turn | 3.10 | 3.10 | 1.99 | 1.21 | 1.10 | 1.04 |

Table 6: Comparison of dialogue context graphs built by local semantics and global semantics.

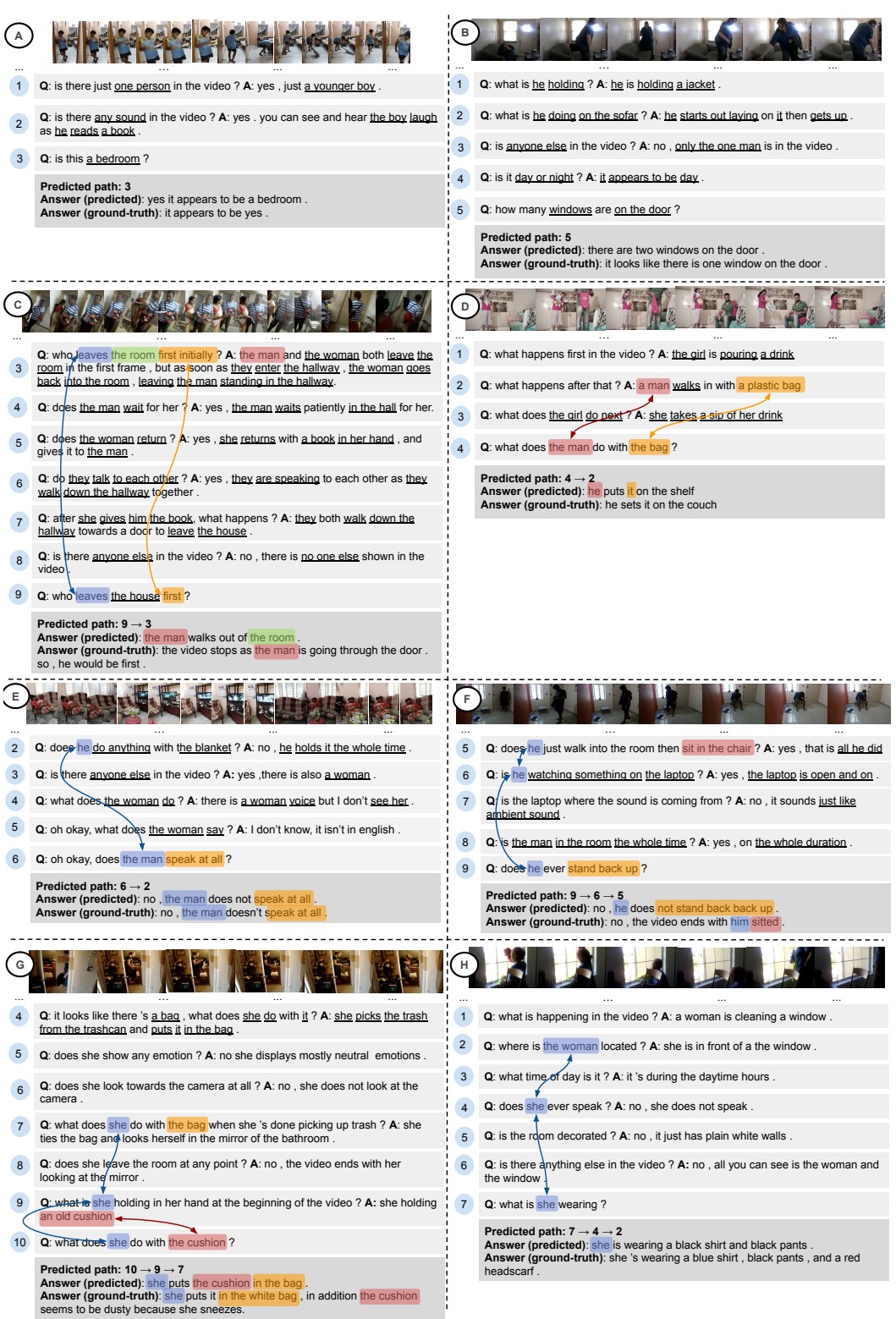

Figure 4: Example outputs of reasoning paths and dialogue responses.

