# OpenReview forum: "Learning Reasoning Paths over Semantic Graphs for Video-grounded Dialogues"
_ICLR.cc/2021/Conference — ICLR 2021 Poster_

### Official Review · AnonReviewer2 · 2020-10-27
**Interesting engineering contribution, but the underlying principle seems not really new and lack of discussion with relevant related works**

**Rating:** 7
**Confidence:** 4

**Review:**

Summary:

This paper addresses the visual question answering in a multi-turn or conversational setting. Given a video (series of frames or images), a model has to reason across space and time to arrive at a correct answer for a given question. This task involves understanding the content and context of dialogue turns, i.e., given a question and N dialogue turns, only M<<N of the dialogue turns are strongly related to the question posed. This paper proposes to simulate the dependencies between dialogue turns, forming a reasoning path, to answer a given question. In a way, the proposed approach selects relevant dialogue turns that are useful to answer the question.

There are two steps to make the reasoning path:
(1) At each dialogue turn, a graph network is constructed at the turn level. Any two turns are connected if they have an overlapping lexical span or if their lexical spans are semantically similar.
(2) Secondly, a path generator is trained to predict a path from the current dialogue turn to past dialogue turns that provide additional and relevant cues to answer the current question.

Ultimately, the main idea to create a reasoning path is based on compositional semantic similarities.



Comments (Technical, Major Flaws of this paper):

(A) I am not sure whether the author(s) is aware, but from the NLP perspective, the current method (step 1) is trying to simulate the discourse structure of dialogues. I believe that this is an important direction, and the uniqueness of this works lies in the multi-modality of the input, i.e., possibility of the interplay between texts and images (using the information in both modalities).
- The claimed novelty in this paper is in the construction or usage of reasoning graph, i.e., to construct a graph structure to connect turn-level representations in dialogue. However, in Step 1, the use of entity and/or compositional similarity to create a graph structure out of a text is not new at all, and the paper fails to cite related works, as if it is the first one to propose this. In fact, the idea has been used in NLP for a long time (albeit mostly in the monologue).
- I am not sure whether combining entity with action phrases (called "lexical spans" in the paper) is new. Can you confirm whether the proposed "lexical spans" is indeed new to construct/simulate the discourse structure?
- Regarding step 1, perhaps the main contribution of this paper is applying the idea to dialogues, instead of monologues? Another possible contribution is "filtering out" unimportant semantic relations. In normal discourse structure, all parts of texts are connected in a single structure. However, in the context of this paper, only edges that are relevant to the posed question are used.
- Unless the paper can discuss the related NLP works for step 1, I can only treat this paper as the extension of the corresponding NLP method in a multi-modal setting. There is an engineering contribution, but not from the methodological (theoretical) perspective.
- I think the author(s) will benefit much by surveying papers on discourse structures (or the ``shallow" construction of them), instead of machine reading comprehension. Many studies tried to establish discourse structure (albeit in a monologue) using entities that are mentioned and their semantic representations. A few of such works are:
  R. Barzilay and M. Lapatta. 2008. Modelling Local Coherence: An Entity-based Approach. https://www.aclweb.org/anthology/J08-1001.pdf
  C. Guinaudeau and M. Strube. 2013. Graph-based Local Coherence Modeling. https://www.aclweb.org/anthology/P13-1010/
  J.W.G. Putra and T. Tokunaga. 2017. http://www.aclweb.org/anthology/W/W17/W17-2410.pdf
- The currently proposed method step 1 seems to be the combination of entity graph + semantic similarity graph in these related works, but the current paper "filters" only edges relevant to the posed question.
- A related work to construct the discourse structure in dialogues:
+ G. Morio and K. Fujita. 2018. End-to-end Argument Mining for Discussion Threads Based on Parallel Constrained Pointer Architecture. https://www.aclweb.org/anthology/W18-5202.pdf

(B) The reasoning model, which is a combination of GCN + transformer can be interesting. However, the idea of cross-modality representation refinement is somewhat similar to what has been studied in VQA.
  Le, T. M., Le, V., Venkatesh, S., & Tran, T. (2020). Dynamic Language Binding in Relational Visual Reasoning. In IJCAI 2020.
  Gao, P., Jiang, Z., You, H., Lu, P., Hoi, S. C., Wang, X., & Li, H. (2019). Dynamic fusion with intra-and inter-modality attention flow for visual question answering. In CVPR 2019.

(C) After constructing the reasoning path (in response to the given question), the next step is to decode such representation to generate the answer. This paper proposes to use the transformer model to do that. I believe the use of the transformer model to generate text is not new. In fact, the author(s) mentions this in the paper (the last paragraph of Section 3.4).

(D) In overall, if we look at the pipeline (system) level, the proposed pipeline is new (the whole process). However, I seriously concern about the step (1) of the proposed method (page 1). My main concern about this paper is its lack of awareness of related works in text processing (step 1 of their method). In fact, it fails to cite relevant works (that are very similar to this work). I might appreciate this paper in terms of engineering contribution (in a multi-modal setting), but I cannot acknowledge that step 1 is novel. Having that said, I think the authors need to provide a comparison to related works, proving the novelty of the current method. I am willing to increase the rating if the authors can properly address my concerns during the rebuttal phase.

(E) The content from 3.3 to 3.4 is very hard to follow.
Correction of terms:
	- linguistic dependency parsers --> "syntactical" dependency parsers (this is the correct term)
	- linguistically, the term "lexical span" is weird. A span is a series of continuous lexicons (in the text surface). I suggest using a better term, as the "lexical span" in this paper might be discontinuous (do I misunderstand?).

---

> ### Author Response · Authors · 2020-11-24
> **Response to Reviewer 2**
>
> We want to thank R2 for the valuable and detailed feedback to improve the paper. Please find below our response. We also revised and updated the paper itself.
>
> * Concern #1:
> “in Step 1, the use of entity and/or compositional similarity to create a graph structure out of a text is not new at all, and the paper fails to cite related works”:
>
> Thank you for your comments about the relevance of our work to this line of research. We carefully review prior work in this research domain and have improved our paper with citation of these papers (Please refer to Section 2).
>
> There has been related work in discourse structures in dialogues. However, they focus on very different types of dialogues (e.g. online discussion forums) for argument mining tasks with very different problem settings from our paper. In our work, the types of dialogues are closer to casual open-domain dialogues with less presence of argument components. Moreover, the presence of video as a common grounding information among dialogue turns also necessitates cross-modality interaction. For more details, we reviewed research of discourses in monologues and dialogues and provided comparison to our approach in Section 2.
>
> Compared to prior work in discourse structures, our modification of lexical spans which combine both entities and action phrases, are designed for more comprehensive cross-modality interaction between text and visual features. In a video, its visual features are not limited to just entity-based information but often contain action-based signals as well. Including action phrases supports the cross-modality interaction process to obtain relevant visual information.
>
> * Concern #2:
> “The idea of cross-modality representation refinement is somewhat similar to what has been studied in VQA.”:
>
> Indeed, cross-modality representation learning is not new and has been studied in VQA problems. Instead, in this paper, we address a different challenge when the problem is positioned in a multi-turn setting. Our approach directly tackles how cross-modality features are passed temporally from turn to turn in a response generation task. Please see Section 2 for our explanation.
>
> * Concern #3:
> “I believe the use of the transformer model to generate text is not new.”:
>
> Indeed, this method has been used before. However, this is not the central contribution of the paper and we described the technical details for completeness of our method. The main argument of our paper focuses on how a dialogue model learns to extract contextual cues turn by turn.
>
> * Concern #4:
> Technical term definitions:
>
> Thank you for your suggestion. We have corrected some of the technical terms in the current manuscript, e.g. changing ‘lexical spans’ to ‘sub-nodes’, and will incorporate all of the feedback in the final version.

---

### Official Review · AnonReviewer4 · 2020-10-27
**Moderate contribution but lacking some details concerning the implementation**

**Rating:** 6
**Confidence:** 5

**Review:**

The paper studies the problem of video-grounded multi-turn QA and adopts reasoning paths to exploit dialogue information.

Sequential: fail to exploit long turn dependencies
Graphical: fixed structure, fail to factor temporal dependencies
The proposed reasoning path method: balanced between sequential and graphical

It first constructs a turn-level semantic graph based on overlapping lexical span:
- Extract lexical spans from each turn (<Q, A> pair) using a (Stanford) parser
- Two turns are connected if one of their corresponding lexical spans are similar (in terms of word2vec embedding).

Then it trains a path generator to predict paths from each turn to its preceding turns:
- It starts from the current turn and auto-regressively finds the most dependent preceding turn with Transformers
- The turn-level semantic graph is used to mask the dependencies.
- It is trained with supervised loss where the target paths are constructed by running BFS on the semantic graph

Finally, the proposed paths are used to employ multimodal reasoning:
- Visual features are combined with turn level attention
- Multi-model turn-level embeddings are propagated using GCN
- Then use SOTA decoder to generate language response

The author conducts experiments on a benchmark, and the proposed method achieves better QA performance than SOTA without a pre-trained language model and achieves comparable performance when the pre-trained language model is involved.

The author further studies different variations of graph structures and show that using graphs constructed based on lexical spans is better than fully connected graphs or graphs based on whole sentence embedding. And it also shows that including bidirectional edges does not necessarily improve the performance.

A nice feature of the method is that the generated reasoning path can serve as extra explanations for the answer.

Some concerns:

- The model is graph-based thus is restricted to scenarios with a small number of turns, and becomes computationally expensive for long-conversation scenarios.
- Need a more detailed explanation of how the message passing part (section 3.4) is trained.
- Each pair of turns may share multiple pairs of lexical spans that are identical, e.g. Figure 3-A, “she” in turn 10, but there are 2 “she”s in turn 9. Does the frequency influence the similarity?
- It would be more convincing if it gives an analysis of failure cases.
- Section 3.3: Eq.(4), what is the initial $D_0$ correspond to $Z_0$?
- The reasoning path generator uses $C_t$ as input, does it include $A_t$ during inference?

Minor concerns:

- Many symbols are used before their definition:
    > Explanation of $\mathcal{V}$ is first given in Algorithm 1 (section 3.2) but first used in Eq.1 (section 3.1).
    > Section 3.3, 2nd paragraph, 4th line: undefined symbols $\hat{r}_1,\dots,\hat{r}_{m-1}$. They are later mentioned as turn indices in section 3.4, last line of page 5.
- Page 5, line 2: “incorporate” —> “incorporates”.
- Index m is used as a word position in Eq.(1) but becomes a decoding step from section 3.3.

---

> ### Author Response · Authors · 2020-11-24
> **Response to Reviewer 4**
>
> We want to thank R4 for the valuable and detailed feedback. We are glad that R4 found our method beneficial as “the generated reasoning path can serve as extra explanations for the answer.” Please find below our response:
>
> * Concern #1:
> “The model is graph-based thus is restricted to scenarios with a small number of turns, and becomes computationally expensive for long-conversation scenarios”:
>
> We agree that the approach involving graph structures requires additional computational effort. To construct graph structures, we recommend to preprocess dialogue utterances cumulatively, i.e. the semantic graph of the current turn is built on top of the graph from the previous turn. In this way, the computation expense is linear as the dialogue length increases. To have an overview of the graph data, in the revised paper, we provided statistical details and analysis of the constructed semantic graphs (Please refer to Appendix D).
>
> * Concern #2:
> “It would be more convincing if it gives an analysis of failure cases.”
>
> Thank you for your suggestion! We added an analysis of scenarios where our models are susceptible to in the Appendix C. There are two main areas we think the current approach can be improved, including cases with long complex utterances and contextualized semantic similarity.
>
> * Concern #3:
> “The reasoning path generator uses $C_t$ as input, does it include $A_t$ during inference?”:
>
> No, it does not use $A_t$ as input during inference. Given a current turn $t$, $C_t$ is limited to questions and answers of past dialogue turns only, i.e. turn $1$ to turn $t-1$.
>
> * Concern #4:
> Questions regarding section 3.3 and 3.4
>
> In section 3.3, Eq (4), the initial $Z_0$ is the embedding of the current turn index $t$. We added this description in the current revision. In section 3.4, we added an explanation of model training.
>
> * Concern #5:
> “ Does the frequency influence the similarity?”:
>
> Currently we do not factor in the frequency of identical lexical spans in an utterance. It is a good observation and we will try to incorporate it in our final version.
>
> * Concern #6:
>  Math symbol definition and typos:
>
> Thank you! We have improved the paper and will thoroughly check again in the final version.

---

### Official Review · AnonReviewer3 · 2020-10-28
**Learned reasoning paths are very interesting but ablation studies show limited benefits of proposed components**

**Rating:** 6
**Confidence:** 4

**Review:**

This paper proposes creating a semantic graph connecting the multiple turns in a dialogue and subsequently learning reasoning paths in that graph to find the most relevant nodes for answering a given question in a dialogue context.

Strengths:
* This paper proposes to learn non-linear information flows in a sequential data which is a well-motivated problem.
* The proposed method is novel. Training a transformer decoder to learn reasoning paths and using BFS supervision to find the ground-truth paths is very interesting (however empirical support that this is effective is lacking, see Weaknesses below).
* The proposed approach significantly and consistently outperforms existing benchmarks on the AVSD dataset.
* The illustration in Fig 3 demonstrates the benefit of using the selected nodes from the graph.

Weaknesses:
* From the results in Table 1, 3, it is clear that multimodal reasoning over "relevant" nodes in the semantic graph (termed as reasoning path in the paper) helps the model by reducing the noise. However, whether there is any benefit to learning that path is unclear from the ablation study present in Table 5 where there is no noticeable difference in the results between learned v/s fixed paths. In fact, "Path through last 7 turns" performs comparable to the "Learned Path" which raises question about the usefulness of transformer decoder based path learner.
* Similarly results in Table 4 seem within noise error from each other, hence making the arguments "component lexical overlap is better than global lexical overlap" weak.
* It is unclear if this approach of creating the graph (by finding pairwise semantic similarity) is scalable to real world datasets containing several turns with potentially large number of tokens. There is no analysis of information redundancy among the nodes of the graph which can help prune the graph.
* The results are only presented on one dataset. The transformer decoder learning seems to be tuned to this particular dataset ("greedy approach works better due to small size of V"). This questions the generality of the proposed approach.

Questions / Suggestions
* It is unclear what (if) makes this method specific to video-grounded Q/A. Can this methodology be applied to other problems involving dialogues? Can this be applied to any problem containing time series data?
* There needs to be some discussion as to why RLM performs better/comparable (Table 1) when using pretraining.

---

> ### Author Response · Authors · 2020-11-24
> **Response to Reviewer 3**
>
> We want to thank R3 for the valuable and detailed feedback. We are glad that R3 found our method novel with demonstrated benefit of using the selected nodes from graphs. Please find below our response:
>
> * Concern #1:
> “whether there is any benefit to learning that path is unclear from the ablation study present in Table 5 where there is no noticeable difference in the results between learned v/s fixed paths”:
>
> In Table 5, we demonstrate that the performance of the learned path outperforms most baseline models exploiting fixed temporal paths. While the performance using ‘Path through last 7 turns’ might be close (except for CIDEr score), our learned path approach does not require a finetuning process to explore which fixed path configuration is optimal in the current dataset (e.g. last 7 turns in the AVSD benchmark).  Our approach enables a model to learn to extract paths instead. Beyond numerical results, our model also improves system transparency as it can explicitly output a clear reasoning path through past temporal steps in dialogue context (Please refer to Figure 3 and 4). One important application that our approach can be used is to assess the complexity of dialogues when creating new benchmarks involving multi-turn reasoning.
>
> * Concern #2:
> “Results in Table 4 seems within noise error from each other, hence making the arguments "component lexical overlap is better than global lexical overlap" weak”:
>
> About the experiment results in Table 4, even though the results between graphs built upon component lexical overlap and global lexical overlap are closed in some metrics (e.g. METEOR and ROUGE-L), the former approach is more robust in cases of long dialogues which involve more entities and action phrases. This method also provides a clearer way to extract automatic supervision of reasoning paths by selecting the best paths with the most number of component lexical overlap (Please refer to Section 3.3, Data Augmentation).
>
> * Concern #3:
> “It is unclear if this approach of creating the graph (by finding pairwise semantic similarity) is scalable to real world datasets containing several turns with potentially large number of tokens”:
>
> About the scalability of our approach, we experimented with the AVSD benchmark which contains dialogues designed to be close to real world dialogues. Each dialogue was created by human annotators and contains up to 10 dialogue turns, and each question/answer in each turn has close to 10 tokens in average (See Table 2).  To have an overview of the graph data, in the revised paper, we provided statistical details and analysis of the constructed semantic graphs (Please refer to Appendix D).
>
> * Concern #4:
> “It is unclear what makes this method specific to video-grounded Q/A. Can this methodology be applied to other problems involving dialogues?”:
>
> Our method can be applied to other problems involving dialogues that often contain multiple dependencies among dialogue turns. We chose to investigate video-grounded dialogue in which each dialogue turn is explicitly grounded in a common source, i.e. video, and hence, dialogue turns are deemed related via this grounding information. In traditional chit-chat or open domain dialogue problems, there might be less number of cases with clear semantic dependencies among turns. An interesting extension of our method to open-domain dialogues is in scenarios in which the dialogues are situated in a multi-modal environment with topical constraints.

---

### Decision · Program_Chairs · 2021-01-07
**Final Decision**

**Decision:**

Accept (Poster)

**Comment:**

This paper studies the problem of visual question answering in multi-turn dialogues.
The proposed method is to identify relevant dialog turns as a path in a semantic graph that connects the dialogue turns. Empirical performance of the proposed method is strong. Reviewers concerns have been compressively addressed. Overall, the paper has novelty, and explores an interesting direction in this line of work.